# School-Based Sleep Education Program for Children: A Cluster Randomized Controlled Trial

**DOI:** 10.3390/healthcare11131853

**Published:** 2023-06-26

**Authors:** Si-Jing Chen, Shirley Xin Li, Ji-Hui Zhang, Siu Ping Lam, Mandy Wai Man Yu, Chi Ching Tsang, Alice Pik Shan Kong, Kate Ching Ching Chan, Albert Martin Li, Yun Kwok Wing, Ngan Yin Chan

**Affiliations:** 1Li Chiu Kong Family Sleep Assessment Unit, Department of Psychiatry, Faculty of Medicine, The Chinese University of Hong Kong, Hong Kong SAR, China; sijingchen@cuhk.edu.hk (S.-J.C.); jihui.zhang@cuhk.edu.hk (J.-H.Z.); joyce@fellowmindfitness.com (S.P.L.); mandyyu@cuhk.edu.hk (M.W.M.Y.); jessietsang@cuhk.edu.hk (C.C.T.); 2Department of Psychology, The University of Hong Kong, Hong Kong SAR, China; shirleyx@hku.hk; 3The State Key Laboratory of Brain and Cognitive Sciences, The University of Hong Kong, Hong Kong SAR, China; 4Center for Sleep and Circadian Medicine, The Affiliated Brain Hospital of Guangzhou Medical University, Guangzhou 510370, China; 5Key Laboratory of Neurogenetics and Channelopathies of Guangdong Province and the Ministry of Education of China, Guangzhou Medical University, Guangzhou 510260, China; 6Department of Medicine and Therapeutics, Faculty of Medicine, The Chinese University of Hong Kong, Hong Kong SAR, China; alicekong@cuhk.edu.hk; 7Department of Paediatrics, Faculty of Medicine, The Chinese University of Hong Kong, Hong Kong SAR, China; katechan@cuhk.edu.hk (K.C.C.C.); albertmli@cuhk.edu.hk (A.M.L.); 8Laboratory for Paediatric Respiratory Research, Li Ka Shing Institute of Health Sciences, Faculty of Medicine, The Chinese University of Hong Kong, Hong Kong SAR, China; 9Hong Kong Hub of Paediatric Excellence, The Chinese University of Hong Kong, Hong Kong SAR, China

**Keywords:** sleep education, children, sleep pattern, parent engagement, behavioral outcome

## Abstract

Insufficient sleep contributes negatively to child developmental processes and neurocognitive abilities, which argues the need for implementing interventions to promote sleep health in children. In this study, we evaluated the effectiveness of a multimodal and multilevel school-based sleep education program in primary school children using a cluster randomized controlled design. Twelve schools were randomly assigned to either the sleep education or nonactive control groups. The sleep education group included a town hall seminar, small class teaching, leaflets, brochures, and a painting competition for children. Parents and teachers were invited to participate in a one-off sleep health workshop. Parental/caregiver-reported questionnaires were collected at baseline and 1-month follow-up. A total of 3769 children were included in the final analysis. There were no significant improvements observed in the sleep-wake patterns, daytime functioning, and insomnia symptoms between the two groups at follow-up, whereas the intervention group had significantly improved parental sleep knowledge than the controls (paternal: adjusted mean difference: 0.95 [95% confidence interval (CI): 0.18 to 1.71]; maternal: adjusted mean difference: 0.87 [95% CI: 0.17 to 1.57]). In addition, children receiving the intervention had a lower persistence rate of excessive beverage intake (adjusted odds ratio: 0.49 [95% CI: 0.33 to 0.73]), and experienced greater reductions in conduct problems (adjusted mean difference: 0.12 [95% CI: 0.01 to 0.24]) compared with the controls at 1-month of follow-up. Moreover, a marginally significant reduction for emotional problems in the intervention group was also observed (adjusted mean difference: 0.16 [95% CI: −0.00 to 0.32]). These findings demonstrated that school-based sleep education was effective in enhancing parental sleep knowledge and improving behavioral outcomes in children, but not sufficient in altering the children’s sleep-wake patterns and sleep problems.

## 1. Introduction

An adequate amount of good sleep is essential for child development, growth, and learning [1]. Thus, insufficient sleep can lead to adverse developmental and neurocognitive outcomes [2,3], and children with insufficient sleep tend to have more externalizing behaviors, such as increased irritability, hyperactivity, and inattention [4,5]. Despite the deleterious impacts of insufficient sleep, children in Asia in fact experienced a secular decline in sleep duration over the past decade [6]. Asian children, in particular Chinese children, tend to have a shorter sleep duration, characterized with a later bedtime and earlier wakeup time compared with those from non-Asian countries, which may be due to culturally based differences in school schedules, homework load, and sleep practices [7,8,9,10]. In addition, the sleep patterns and sleep problems in Asian children have been found to be associated with adverse physical and mental health consequences [11,12,13]. Therefore, there is an urgent need for implementing effective interventions to promote sleep health in this particular group [14,15], perhaps even more so in Hong Kong children, who seem to face a more severe sleep loss due to subcultural differences, as they were found to sleep almost one hour less than their Shanghai peers in a ten-year secular trend of 2003–2012 [9].

The implementation of a sleep health promotion program in a school setting provides a unique opportunity to reach a large proportion of children and their parents. However, previous programs were primarily emphasized on adolescents [16,17,18,19,20]. Nonetheless, researchers have suggested the empirical need to begin sleep health education as early as possible, given the high prevalence of sleep problems and their associated adverse physical and mental health consequences [21,22].

A small body of literature has attempted to provide sleep education for school-aged children with mixed findings, and there is very limited evidence from Asian countries [23,24,25,26,27,28]. Several studies reported that sleep education was associated with improvements in children’s sleep, such as an advanced bedtime and a longer sleep duration [24,25,26,27]. However, two recent studies found no such changes in children’s sleep behaviors after sleep education [23,28]. Nonetheless, several limitations, such as the relatively small sample size [23,24,26,27,28], non-randomized controlled study design [23,24,26,28], lack of active parental involvement [27] and low generalizability (e.g., inclusion of single-year group) [23,24,25,26,28], may lead to uncertainties in the interpretation of the existing evidence. Therefore, current knowledge is not sufficient to draw a conclusion concerning the effects of sleep education on changing children’s sleep behaviors. The current study presented data from a multimodal and multilevel school-based sleep education program among primary school children. We hypothesized that students included in the sleep education program would demonstrate greater improvements in sleep, daytime functioning, and behaviors compared to the control group.

## 2. Materials and Methods

### 2.1. Study and Participants

This study was part of our sleep education program termed “Healthy Sleep, Health School Life” which involved both primary and secondary schools. The details of subject recruitment have been published previously [20]. Fifteen local primary schools agreed to join the current study. Grade one to five students were invited to join the study. Three schools dropped out before intervention, and a total of 12 schools (*n* = 5178) were randomly assigned to the sleep education or the nonactive control groups at a 1:1 ratio (Figure 1). The randomization of the current study was based on a single sequence of random assignments to individual schools as a cluster. One of the authors (M.W.M.Y.) was responsible for the randomization process, including generating the random allocation sequence, enrolling the school, and assigning a school to the intervention or the control groups. Ethical approval was granted by the clinical research ethics committee (CRE-2011.249-T), and the trial was registered with the Chinese Clinical Trial Registry (ChiCTR-TRC-12002798).

### 2.2. Sample Size Estimation

Previous studies have shown that the effect size of school-based intervention for improving sleep duration was between 0.03 and 0.10, respectively [19,29]. In order to achieve a power of 0.90 while allowing a type I error of 0.05, a power analysis based on a medium effect size (0.07) suggested that a total of 1293 subjects per group is required. On the assumption that 70% of the students would agree to take part in the study, a total of 3694 participants would be required (with 1847 subjects in each group).

### 2.3. Procedure

Following the approval from the school principals and teachers, a set of baseline questionnaires packed in coded envelopes were delivered to the school children. The questionnaires were completed by either the parents or caregivers who were required to return the package of the materials within two weeks, and only those who provided parental consent were enrolled in the final analysis, even though all students in the intervention schools received sleep education.

### 2.4. Intervention

A series of educational workshops, including a town hall seminar, two small interactive class teachings, and a painting competition were held for schools in the intervention group over a period of three months during designated class time. The town hall seminar lasted for 45 min and were delivered by the co-authors (Y.K.W., A.M.L., A.P.S., and S.P.L.) who were clinicians and researchers in sleep medicine. The interactive classes lasted around 35–40 min depending on the school’s usual class schedule and were delivered by research assistants who received two training sessions on sleep medicine together with one practical mock session supervised by the co-authors (Y.K.W and S.P.L).

The program in the current study mainly covered the following topics: (1) sleep facts, (2) the importance of sleep, (3) barriers to adequate sleep, (4) good sleep hygiene practice, and (5) consequences of sleep deprivation. The details of the program contents were presented in the previously published study with a modification to match the developmental needs and inclusion of age-appropriate content for school children, such as storytelling, case studies, and group activities [20]. The town hall seminar involved didactic teaching, which emphasized on the importance of sleep and the consequences of sleep deprivation, while the small class workshop provided students with examples of good sleep hygiene practice, time and stress management. The program also included behavioral practice by providing students with a goal-setting sheet to set an ideal wakeup time and bedtime to achieve an optimal sleep amount. Moreover, homework that requested the involvement of their parents was also included. Students were encouraged to surf the sleep-health website with the help of their parents to complete the worksheet and invite their parents to guide their pre-set behavioral practice. In addition, leaflets and brochures were also offered to students and parents as additional materials.

Parents and teachers were invited to attend a one-off workshop to equip them with proper sleep knowledge and sleep practice. The initial assessments were conducted from late December 2011 to early February 2012. The educational program was delivered between February to May 2012, and the follow-up assessments were conducted about one month after the end of the intervention.

### 2.5. Measurements

The Hong Kong children sleep questionnaire (HKCSQ) was used to measure sociodemographic and sleep-related characteristics, such as insomnia symptoms and sleep-wake patterns on both weekdays and weekends, and lifestyle practice including tea, coffee, energy drinks, and beverage consumption in the past month [20,30]. Insomnia symptoms, including difficulty initiating sleep (DIS), difficulty maintaining sleep (DMS), and early morning awakening (EMA) were assessed, and frequent insomnia symptoms were defined as having either one of these symptoms at least three times per week in the past month. Sleep duration was reflected by the time-in-bed, and was calculated as the difference between the bedtime and wakeup time during the school days and weekends [31]. Lifestyle practices, such as caffeinated consumption (tea, coffee, energy drink, and soft drink) was defined as abnormal if it occurred at least three times per week in the past month.

Children’s medical and psychiatric illnesses were measured with questions asking whether the participants had been diagnosed with any of the following chronic diseases: cancer, eczema, asthma, psychiatric/mood disorders, chronic pain, and gastro-esophageal reflux disease.

The pediatric daytime sleepiness scale [32] was used to measure children’s daytime sleepiness using 8-item which described the sleep-related behaviors on a 5-point Likert scale. It has been validated in Chinese children who demonstrated acceptable psychometric properties (test-retest reliability: r = 0.78; internal consistency: Cronbach’s alpha = 0.66) [33]. A higher score indicated a greater daytime sleepiness.

The parent-reported strengths and difficulties questionnaire, which has been validated in Chinese primary school children (test-retest reliability of all the subscales and the total difficulty: r ranges from 0.75 to 0.86; internal consistency: Cronbach’s alpha ranges from 0.45 to 0.81), was used to assess children’s behaviors in the following five aspects: emotional problems, conduct problems, peer relationships, attention/hyperactivity, and prosocial behaviors [34]. The less internally consistent subscales were emotional problems (alpha = 0.66), conduct problems (alpha = 0.65), and peer relationships (alpha = 0.45). 

Parental sleep knowledge was measured by the Chinese version of the sleep knowledge questionnaire, which is a self-administered questionnaire that consists of 15 items, and has been validated in the general population (test-retest reliability: r = 0.90; internal consistency: Cronbach’s alpha = 0.65), with a total score ranging from –30 to 30 [35].

### 2.6. Data Analyzes

Analyzes were conducted based on the intent-to-treat approach. The primary outcomes were sleep-wake patterns. Chi-square test tests for categorical variables and t-tests for continuous variables were performed to examine the differences in the baseline sociodemographic characteristics between the two groups. The intervention effects on dichotomous variables (e.g., insomnia symptoms) and continuous variables (e.g., sleep-wake patterns) were analyzed using generalized estimating equations (GEE) logistic regression model and linear mixed-effects model (LMM) analysis, adjusting for the clustering effect and baseline characteristics, respectively. Multiple imputations were used to address missing data in the baseline characteristics. In GEE analyzes, an adjusted odds ratio with a 95% confidence interval (CI) was used to investigate the associations of the intervention with insomnia symptoms and lifestyle practice changes. Subjects were divided into two subgroups based on their baseline responses to explore the incidence and persistence rates for insomnia symptoms and lifestyle practices. Adjusted between-group differences in the estimated mean change from the baseline to each time point were used to calculate effect sizes in LMM analysis. Statistical analyzes were performed using IBM Statistical Package System Software (SPSS) 26.0 and Stata 17.0. A *p*-value of <0.05 was considered statistically significant.

## 3. Results

### 3.1. Demographics

At baseline, 3806 children (response rate: 73.5%; intervention group: 2094 students; control group: 1712 students) returned the questionnaires. Participants were excluded if they were not between the ages of 6 and 12 years (*n* = 37). Thus, a total of 3769 children and their parents (intervention group: 2086 students, 1915 mothers and 1734 fathers; control group: 1683 students, 1538 mothers and 1411 fathers; 49.2% girls; mean ± SD age: 8.8 ± 1.5 years) were included in the final analysis (Figure 1). Among them, 2863 participants (response rate: 76.0%; intervention group: 1677 students; control group: 1186 students) returned valid follow-up questionnaires with complete data regarding the main measurements. Participants who completed the follow-up were younger and more likely to be girls, and their parents had a lower education background and family income at baseline (all *Ps* < 0.05), but they were similar to non-completers in terms of parental employment as well as the children’s medical and psychiatric conditions. In addition, they also adopted earlier weekday and weekend bedtimes, an earlier weekend wakeup time, and a longer weekday duration at baseline compared to the non-completers (all *Ps* < 0.05). Most of the baseline (74.6%) and follow-up questionnaires (73.5%) were filled out by the mothers, and about one-fifth of the questionnaires were filled out by the fathers (baseline: 19.8%; follow-up: 22.1%). The rest of the questionnaires were completed either by both parents (baseline: 0.8%; follow-up: 0.3%) or caregivers (baseline: 4.8%; follow-up: 4.1%).

There were no differences observed in sex distribution between the intervention and control groups (Table 1). However, the intervention group was older, and had fewer medical and psychiatric illnesses, along with a lower parental education background, parental employment rates, and monthly household income compared with the control group (all *Ps* < 0.05). Thus, the final analyzes were adjusted for all these variables. In terms of sleep-wake patterns, the two groups had comparable weekend wakeup times and sleep durations, whereas the intervention group had earlier weekday and weekend bedtimes, later weekday wakeup times, and longer weekday sleep durations compared to the controls at baseline (Table 2).

The average weekday and weekend sleep duration was 8 h 56 min and 10 h 13 min, respectively. Approximately half (45.0%) of children had less than 9 h of sleep during the weekdays, and over 40% of them had a subjective perception of sleep insufficiency.

### 3.2. Sleep-Wake Patterns

Table 2 presents the data on the children’s sleep-wake patterns, daytime sleepiness, behavior problems, and parental sleep knowledge level. There were no significant between-group differences observed in the changes in the weekday bedtime (adjusted mean difference: −0:00 [95% CI: −0:02 to 0:02], *p* = 0.90), wakeup time (adjusted mean difference: −0:00 [95% CI: −0:01 to 0:01], *p* = 0.74) and sleep duration (adjusted mean difference: −0:00 [95% CI: −0:01 to 0:01], *p* = 0.86) from baseline to follow-up. Meanwhile, for weekend sleep practice, there were significant differences observed in the changes of the weekend bedtime, with children in the intervention group being found to have established a later bedtime at follow-up (adjusted mean difference: −0:03 [95% CI: −0:06 to −0:00], *p* = 0.01). However, for the mean changes in the weekend wakeup time (adjusted mean difference: −0:02 [95% CI: −0:08 to 0:03], *p* = 0.42) and sleep duration (adjusted mean difference: −0:00 [95% CI: −0:07 to 0:06], *p* = 0.81), no significant differences were found between the two groups.

### 3.3. Children Daytime Sleepiness and Behavioral Outcomes and Parental Sleep Knowledge

No significant differences were recorded in the mean changes of the daytime sleepiness score from baseline to follow-up between the two groups. For the strength and difficulties scale, children who received sleep education reported improved conduct problems (adjusted mean difference: 0.12 [95% CI: 0.01 to 0.24], *p* = 0.03), and had fewer emotional problems at a marginally significant level (adjusted mean difference: 0.16 [95% CI: −0.00 to 0.32], *p* = 0.05) compared to the control group at follow-up, but they had a lower degree of improvements in prosocial behavior compared with the controls (adjusted mean difference: −0.19 [95% CI: −0.31 to −0.07], *p* = 0.002). In terms of parental sleep knowledge, parents in the intervention group were found to have a significantly improved sleep knowledge compared to the control group at follow-up (paternal: adjusted mean difference: 0.95 [95% CI: 0.18 to 1.71], *p* = 0.02; maternal: adjusted mean difference: 0.87 [95% CI: 0.17 to 1.57], *p* = 0.02).

### 3.4. Insomnia Symptoms and Caffeinated Consumption Practice

The incidence and persistence rates of insomnia symptoms and caffeinated consumption practice were evaluated, as shown in Table 3. The persistence rate of beverage consumption was lower in the intervention group [intervention vs. control: 33.0% vs. 50.8%; adjusted odds ratio: 0.49 [95% CI: 0.33 to 0.73])]. The incidence and persistence rates of other caffeinated products and insomnia symptoms between the two groups at follow-up were found to have no significant differences (all *Ps* > 0.05).

## 4. Discussion

In this large-scale, school-based sleep education program targeting primary school children, we found that the intervention did not show significant effects on the children’s sleep-wake patterns, sleep-related difficulties, and the total difficulties score as reported by their parents. Nonetheless, it led to improvements in parental sleep knowledge and in several behavioral outcomes, including the amelioration of excessive beverage intake, conduct, and emotional problems. The findings of the current study have implications for future sleep intervention programs with respect to the inclusion of motivational strategies, active parental involvement, and a more specific focus.

The increase in parental sleep knowledge after sleep education was found to be consistent with previous studies targeting preschool and school-aged children [26,36,37]. Notably, a recent pragmatic and prospective stepped-wedge cluster trial on the sleep health education program also resulted in improvements in parental knowledge but found clinically insignificant improvements in the sleep outcomes [37]. It seems that the increase in parental knowledge alone is not sufficient to result in improved children’s practice [38]. This discrepancy may be partly due to the gap between parental perceptions and actual sleep requirements in children [37,39,40], as parents tend to underestimate the sleep needs and the presence of sleep difficulties in their children [38,39]. In addition, even though parents and teachers were provided with sleep health workshops in the current study design, parental involvement was not compulsory, and less than one-third of the parents in the intervention group (mother: 30.2%; father: 19.1%) reported their engagement in the program, whereas the degree of their engagement was not documented. Thus, a more active involvement of parents with the inclusion of practical strategies (e.g., setting a bedtime with children, being role models, and providing emotional support) might potentially improve children’s sleep [41,42,43].

The insignificant intervention effects observed in the children’s sleep-wake patterns and sleep difficulties in the current study were somehow inconsistent with previous children sleep interventions which demonstrated a significant improvement in both subjective and objective sleep measures after the intervention [24,25,26,27]. This inconsistency might be due to the differences in methodologies, as previous studies were of a relatively small sample size [24,26,27,28], narrower age group [24,25,26,28], non-randomized controlled study design [24,26,28] and targeting at-risk children (e.g., children with sleep disturbance) or clinical samples [25,27], which may all increase the intervention effects [37,44]. A recent review has overall concluded that sleep intervention in healthy children, albeit with an average of a 10.5 min longer sleep duration, has very small effects and a limited clinical implication [45]. In particular, studies with a specific focus on earlier bedtimes tended to report greater improvements in sleep duration [45], whereas advancing bedtime was not emphasized in the current study. Moreover, the parent-reported approach may be one of the reasons that led to non-significant improvements being observed, as parents might not reflect their children’s actual sleep [24]. Alternatively, there may be a delayed intervention effect as children at a young age may need a longer time to change their behaviors [46]. Nevertheless, our study findings were partly concordant with previous large-scale school-based sleep education targeting adolescents, which demonstrated that education alone is not sufficient to change the sleep behaviors [16,18,19,20,47,48]. Thus, future school-based sleep intervention programs may need to consider other contributing factors, such as motivation and self-efficacy, which are core components in the behavioral changes model [16,29]. Additionally, sleep interventions that have a broader focus were seemingly less effective in improving sleep, suggesting a need to further refine the sleep education content with a specific focus (e.g., advance bedtime) [45]. Furthermore, other school-based interventions, such as delaying the school start time should also be considered as a potential population-wide strategy to address the childhood sleep insufficiency problem [49,50,51,52,53], albeit more evidence is needed.

There were significant improvements observed in several behavioral outcomes, including excessive beverage intake, conduct and emotional problems, which extended our previous findings on adolescents, further supporting the notions that several specific sleep hygiene components might be more penetrating to the children and adolescent populations [20]. Moreover, these improvements could also be attributed to other educational components employed in our program, such as time and stress management skills, which are effective in enhancing the coping skills and reducing stress symptoms that could further lead to the promotion of mental health and reduction of beverage consumption [54,55,56]. In addition, the involvement of significant others (parents and teachers) in the current program may have had some positive effects on shaping the children’s daytime behaviors [57,58], though we did not document the degree of their engagement.

The current study has several strengths. First, this study has a large sample size and employed a cluster randomized control design which ensures adequate statistical power. Second, parents and teachers were involved in the current study. Third, we included a multilevel sleep education program. However, there are several limitations to the current study that need to be acknowledged. First, our study was limited by using parental/caregiver reports instead of objective sleep measurements. Second, the significant findings in parental knowledge, children’s conduct and emotional problems need to be interpreted with caution, as these variables were measured using questionnaires with less satisfactory internal consistency (Cronbach’s alpha < 0.7). Future school-based studies may need to consider more rigorous sleep and behavioral measurements. Third, baseline imbalances, including age, children’s medical and psychiatric illness, parent education and employment, and family income were found between the randomized groups. Thus, we adjusted for these baseline characteristics with significant between-group differences in the analyses. Moreover, children’s sleep and behaviors were assessed only one month after intervention, and longer follow-ups may be necessary to detect any possible long-term intervention effects as behavioral changes often take a longer time [46], particularly among young children. Lastly, the involvement of significant others, and potential factors, including the motivation to change, time management, and coping skills, might have potentially mediated the improvements observed in the behavioral and emotional outcomes, which were not assessed in this study. Thus, the underlying mechanism of such positive changes cannot be determined in the current study.

## 5. Implications

The findings of the current study add to the existing literature that school-based sleep education is insufficient in changing children’s actual sleep practice, at least from a parental perspective. Even if parents had increased sleep knowledge, their understanding of children’s sleep requirements and their practices may still be inadequate [37,39,40]. Thus, future studies should consider including a more active parental involvement [41,42,43], and further developing sleep education with a specific focus on advancing bedtime. In addition, it is also necessary to develop sleep education programs with the consideration of the behavioral change intervention (e.g., motivational interviewing), which aims to enhance one’s motivation to change by setting goals and providing feedback to modify sleep practices throughout the intervention [16,29].

## 6. Conclusions

The current study demonstrated that multimodal and multilevel school-based sleep education was effective in enhancing parental sleep knowledge and improving several behavioral outcomes in children but was not sufficient in changing the children’s sleep-wake patterns and sleep disturbances. Future sleep intervention programs with longer follow-up durations should consider adding behavioral motivational strategies, a more specific focus, and the systemic involvement of parents to improve children’s sleep.

## Figures and Tables

**Figure 1 healthcare-11-01853-f001:**
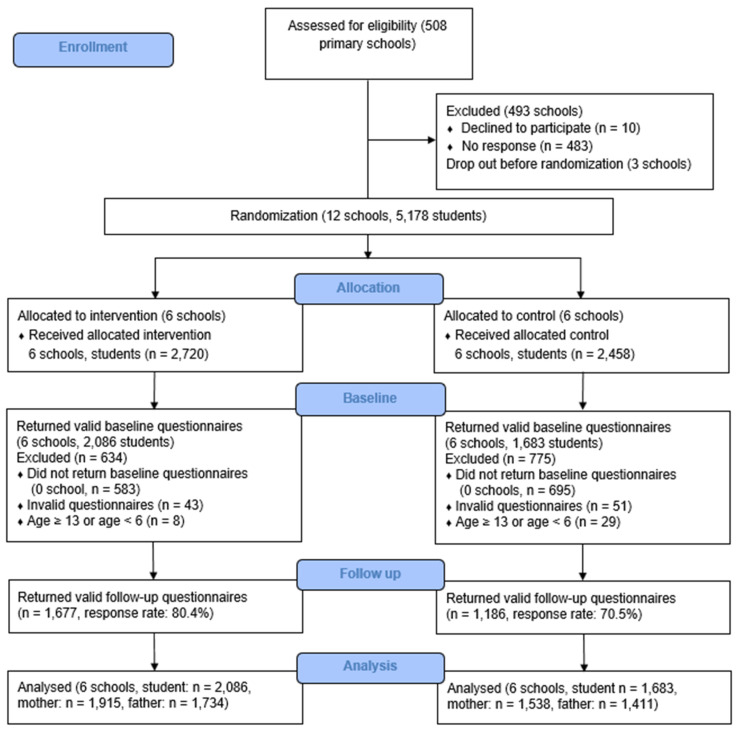
Flow chart of the included schools and the subject recruitment process.

**Table 1 healthcare-11-01853-t001:** Sociodemographic characteristics of subjects in the intervention and control groups.

Characteristic	Control	Intervention	*p*-Value
Age, years, mean (SD) (*n* = 3767)	8.7 (1.5)	8.8 (1.5)	0.02
Sex, female (%) (*n* = 3769)	48.8	49.6	0.61
Medical and psychiatric illnesses (%) (*n* = 3749)	14.7	10.6	<0.001
Paternal education (*n* = 2849)			
Primary or below (%)	3.5	7.0	<0.001
Secondary (%)	50.3	71.8	
Tertiary or above (%)	46.2	21.2	
Paternal employment (%) (*n* = 2897)	94.3	92.1	<0.001
Maternal education (*n* = 3194)			
Primary or below (%)	3.0	7.8	<0.001
Secondary (%)	60.7	74.1	
Tertiary or above (%)	36.3	18.1	
Maternal employment (*n* = 3257)	64.5	57.4	<0.001
Family income (%) (HK$ ≥ 15,000) (*n* = 3554)	78.6	55.6	<0.001

Medical and psychiatric illnesses included cancer, eczema, asthma, psychiatric/mood disorders, and chronic pain or gastro-esophageal reflux disease. SE, standard error.

**Table 2 healthcare-11-01853-t002:** Children sleep/wake pattern, daytime sleepiness and behavioral profile, and parental sleep knowledge at baseline and follow-up.

	Intervention		Control		Unadjusted Analysis ^a^		Adjusted Analysis ^a,b^	
	Baseline ^c^	Follow-Up ^c^	Baseline ^c^	Follow-Up ^c^	Intervention Effect (95% CI) ^d^	*p*-Value	Intervention Effect (95% CI) ^d^	*p*-Value
**Weekdays (*n* = 3765)**								
Bedtime	22:01(0:02) ^e^	22:07 (0:02)	22:07 (0:04)	22:12 (0:04)	−0:00 (−0:02 to 0:02)	0.84	−0:00 (−0:02 to 0:02)	0.90
Wakeup time	7:01 (0:07) ^e^	7:01 (0:07)	6:59 (0:08)	6:59 (0:08)	0:00 (−0:01 to 0:01)	0.75	0:00 (−0:01 to 0:01)	0.74
Sleep duration	9:00 (0:05) ^e^	8:54 (0:05)	8:52 (0:04)	8:47 (0:04)	−0:00 (−0:01 to 0:01)	0.70	−0:00 (−0:01 to 0:01)	0.86
**Weekends (*n* = 3763)**								
Bedtime	22:44 (0:01) ^e^	22:48 (0:01)	22:47 (0:02)	22:48 (0:02)	−0:03 (−0:06 to −0:00)	0.01 *	−0:03 (−0:06 to −0:00)	0.01 *
Wakeup time	8:59 (0:03)	8:52 (0:02)	8:59 (0:05)	8:50 (0:03)	−0:02 (−0:08 to 0:03)	0.38	−0:02 (−0:08 to 0:03)	0.42
Sleep duration	10:15 (0:02)	10:04 (0:01)	10:11 (0:04)	10:01 (0:03)	−0:00(−0:07 to 0:06)	0.82	−0:00 (−0:07 to 0:06)	0.81
**PDSS (*n* = 3704)**	13.55 (0.24) ^e^	12.98 (0.29)	13.98 (0.17)	13.58 (0.25)	0.17 (−0.15 to 0.48)	0.31	0.15 (−0.17 to 0.47)	0.35
**Strengths and difficulties (*n* = 3732)**								
Total difficulty	11.81 (0.46) ^e^	11.36 (0.42)	10.96 (0.35)	10.87 (0.45)	0.36 (−0.03 to 0.75)	0.07	0.35 (−0.03 to 0.72)	0.07
Peer	2.69 (0.15) ^e^	2.68 (0.16)	2.40(0.17)	2.44 (0.15)	−0.05 (−0.19 to 0.09)	0.50	−0.05 (−0.18 to 0.08)	0.48
Emotion	2.49 (0.08) ^e^	2.36 (0.05)	2.33 (0.04)	2.36 (0.09)	0.16 (0.00 to 0.32)	0.049 *	0.16 (−0.00 to 0.32)	0.05
Conduct	2.18 (0.12) ^e^	2.02 (0.11)	1.99 (0.07)	1.96 (0.11)	0.13 (0.01 to 0.24)	0.03 *	0.12 (0.01 to 0.24)	0.03 *
Hyperactive	4.46 (0.13) ^e^	4.30 (0.12)	4.25 (0.10)	4.11 (0.12)	0.03 (−0.12 to 0.17)	0.72	0.02 (−0.12 to 0.16)	0.78
Prosocial ^f^	6.68 (0.15)	6.66 (0.16)	6.75 (0.18)	6.92 (0.16)	−0.19 (−0.31 to −0.07)	0.001 *	−0.19 (−0.31 to −0.07)	0.002 *
**Paternal sleep knowledge (*n* = 3145)**	6.65 (0.76) ^e^	8.32 (0.96)	8.71 (0.95)	9.41 (0.88)	0.96 (0.26 to 1.66)	0.007 *	0.95 (0.18 to 1.71)	0.02 *
**Maternal sleep knowledge (*n* = 3453)**	7.08 (0.94) ^e^	9.12 (0.86)	8.96 (0.89)	10.11 (1.02)	0.89 (0.15 to 1.63)	0.02 *	0.87 (0.17 to 1.57)	0.02 *

CI, confidence interval; and PDSS, pediatric daytime sleepiness scale. ^a^ Model was controlled for the clustering effect. ^b^ Model was adjusted for the clustering effect and baseline characteristics with significant between-group differences, including age, medical and psychiatric illnesses, parental education and employment, and family income. Multiple imputations were used to address missing data in covariates. ^c^ Data were presented as estimated marginal means (standard error). Standard errors were controlled for the clustering effect. ^d^ Intervention effect is presented as the adjusted between-group difference in the mean change from baseline to follow-up. A positive value indicates an effect size in the hypothesized direction. ^e^ Significant differences between groups at baseline (*p* < 0.05). ^f^ Higher scores on the prosocial behavior subscale reflect more strength. * *p* < 0.05.

**Table 3 healthcare-11-01853-t003:** Incidence and persistence rate for insomnia symptoms and caffeinated consumption practice.

	Incidence					Persistence				
	N (%)	Control (%)	Intervention (%)	Crude OR ^a^ (95% CI)	Adjusted OR ^a,b^ (95% CI)	N (%)	Control (%)	Intervention (%)	Crude OR ^a^ (95% CI)	Adjusted OR ^a,b^ (95% CI)
**Insomnia symptoms (≥3/week)**										
DIS	47 (1.7)	1.3	2.0	1.35 (0.60 to 3.04)	1.18 (0.55 to 2.51)	22 (23.7)	25.0	22.8	0.90 (0.45 to 1.79)	0.70 (0.28 to 1.74)
DMS	25 (0.9)	0.9	0.9	1.02 (0.45 to 2.32)	1.04 (0.55 to 1.98)	3 (12.0)	0	17.6	NS ^c^	–
EMA	24 (0.9)	0.4	1.2	1.95 (0.50 to 7.68)	2.10 (0.67 to 6.60)	3 (13.0)	0	18.8	NS ^c^	–
Any symptoms	80 (3.0)	2.2	3.5	1.33 (0.69 to 2.58)	1.29 (0.72 to 2.32)	34 (27.9)	27.1	28.4	1.05 (0.50 to 2.20)	0.85 (0.38 to 1.93)
**Caffeinated consumption (≥3/week)**										
Tea	50 (1.8)	1.6	2.0	1.03 (0.42 to 2.52)	0.89 (0.54 to 1.47)	13 (28.3)	23.1	30.3	1.48 (0.54 to 4.06)	– ^d^
Coffee	14 (0.5)	0.6	0.4	0.58 (0.16 to 2.11)	0.48 (0.15 to 1.51)	1 (33.3)	33.3	0	NS ^c^	–
Energy drink	43 (1.6)	1.6	1.6	0.79 (0.29 to 2.17)	0.72 (0.34 to 1.55)	10 (41.7)	33.3	44.4	1.46 (0.15 to14.28)	– ^d^
Beverage	142 (5.4)	3.7	6.6	1.43 (0.71 to 2.90)	1.35 (0.88 to 2.06)	64 (40.3)	50.8	33.0	0.48 (0.32 to 0.70) *	0.49 (0.33 to 0.73) *

Three or more times per week was defined as abnormal. CI, confidence interval; DIS, difficulty initiating sleep; DMS, difficulty maintaining sleep; EMA, early morning awakening; and OR, odds ratio. ^a^ Model was controlled for the clustering effect. ^b^ Model was adjusted for the clustering effect and statistically different baseline characteristics, including age, medical and psychiatric illnesses, parental education and employment, and family income. Multiple imputations were used to address missing data in covariates. ^c^ Fisher’s exact test was used as 3 cells have an expected count of less than 5 (*p* > 0.05). ^d^ Convergence criteria were not satisfied. * *p* < 0.05.

## Data Availability

The data will be made available upon publication to researchers who provide a methodologically sound proposal for use in achieving the goals of the approved proposal. Proposals should be submitted to the corresponding author.

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
