# Peer review of "School-Based Sleep Education Program for Children: A Cluster Randomized Controlled Trial"

_healthcare, 2023, doi:10.3390/healthcare11131853_

Round 1

Reviewer 1 Report

I like the project very much - the goal, idea, and methodology including the educational program. I believe that the research problem concerning the sleep of children and adolescents and the consequences of the lack of good sleep practices for physical and mental health is extremely important. Dealing with serious mental problems, whether in research or in practice, we sometimes forget how important it is to properly satisfy basic physiological needs for development and health. The size of the study group and multilevel school-based sleep education deserve additional appreciation.

The weaker points of the manuscript presented for review include the psychometric properties of some tools - the Pediatric Daytime Sleepiness Scale has Cronbach's alpha = 0.66, and in the case of the Parent-reported Strengths and Difficulties Questionnaire, a range of reliability indicators for subscales is given, suggesting a low score for some of them. More precise information would be important - what is the reliability index of particular subscales and drawing conclusions carefully on the basis of the results obtained with their use. In the case of the last tool - Parental sleep knowledge, there is no information about psychometric properties at all - adding information is needed.

In the Discussion section, it would be important to refer to which of the significant results were obtained using scales with weaker Cornbach's alpha indicators.

It would also be important to discuss the inconsistencies more thoroughly - no change in sleep-wake patterns, yet parents noticed changes in behavior and emotional regulation. So what was the mediating mechanism? Or were there other uncontrollable factors responsible for improving functioning? The Authors referred to this issue, but it is a surprising result and requires a deeper analysis.

Reviewer 2 Report

The authors present an extremely important study. The overall manuscript is very written. However, there are minor concerns that needs to be addressed.

1. In introduction, the authors mention:  'In particular, Asian children, especially Chinese, tend to have a shorter sleep duration, characterized by later bedtime and earlier wakeup time compared with those from non-Asian countries...please elaborate on this.

Please include why Asian children in particular tends to have poor sleep practise. Do provide any data that looks into the long-term consequence of poor sleep quality among Asian children

2. Please describe how the sample size was measured in detail. Any post-hoc power analysis performed?

3. Please elaborate on the limitation and add clear future recommendation

none
